# Retrieval of convective available potential energy from INSAT-3D measurements: comparison with radiosonde data and its spatial-temporal variations

*Uriya Veerendra Murali Krishna[1], Subrata Kumar Das[1*], Kizhathur Narasimhan Uma[2], and Govindan Pandithurai[1]*

[1]Indian Institute of Tropical Meteorology, Pune-411008, India

[2] Space Physics Laboratory, Vikram Sarabhai Space Centre, Trivandrum-695022, India

[*]Correspondence to Subrata Kumar Das (skd_ncu@yahoo.com)

**Abstract:** Convective available potential energy (CAPE) is a measure of the amount of energy
available for convection in the atmosphere. The satellite-derived data over the ocean and land is used
for a better understanding of the atmospheric stability indices. In this work, an attempt is made for the
first time to estimate CAPE from high spatial and temporal resolution measurements of the INSAT-3D
over the Indian region. The estimated CAPE from the INSAT-3D is comprehensively evaluated using
radiosonde derived CAPE and ERA-Interim CAPE. The evaluation shows that the INSAT-3D CAPE
reasonably correlated with the radiosonde derived CAPE; however, the magnitude of CAPE shows
higher values. Further, the distribution of CAPE is studied for different instability conditions (different
range of CAPE values) during different seasons over the Indian region. In addition, the diurnal and
seasonal variability in CAPE is also investigated at different geographical locations to understand the
spatial variability with respect to different terrains.

**Keywords:** CAPE; INSAT-3D; Monsoon; Diurnal; Instability



## 1. Introduction

The interaction between convection, clouds radiation, and large-scale circulation remains a major source of uncertainty in understanding the climate and climate change. Convection plays a crucial role in the formation of clouds (cumulonimbus). The convective activity prevailing over the atmosphere is the feeding mechanism for the development of weather systems such as thunderstorms and cyclones. Deep convection is one of the usual phenomena in the tropical region which requires three ingredients: instability, moisture, and uplift. Convective available potential energy (CAPE; Moncrieff and Miller, 1976) is a measure of convective potential in the atmosphere that incorporates the instability and moisture ingredients (Johns and Doswell, 1992). In a physical sense, it is the energy available for the free lifting of the air parcel from the level of free convection to the level of neutral buoyancy. CAPE is also the measure of maximum kinetic energy per unit mass of air parcel achievable by convection of moist air (Murugavel et al., 2012). So it can also be used as an estimator of maximum possible updraft velocity.

Climatology of CAPE provides valuable information for severe weather forecasting. Comparing this climatology to near real-time observations from meteorological sensors on satellites could provide valuable information in assessing the risk of severe weather (Breznitz 1984; Golden and Adams 2000; Doswell 2004; Barnes et al., 2007; Rothfusz et al., 2014; Cintineo et al., 2014). For example, high value of CAPE is an indicator of deep convection (Bhat et al., 1996). Johns and Doswell (1992) showed that the largest hail sizes in convection are related to CAPE. In severe weather conditions, CAPE is one of the key indices determining the occurrence of thunderstorms and tornadoes (McNulty, 1995). Further, Williams and Renno (1993) and Dutta and De (1999) have shown that the isolated heavy rainfall events



on any isolated day result from a higher value of CAPE. Craven (2000) found that higher CAPE values and steep lapse rates were ideal for supercell storm and tornado formation. Williams et al. (2002) suggested that CAPE can be used as a predictor of electrification/lightning intensity in deep tropical convection. The changes in convective activity and atmospheric energy budget are associated with the long-term changes in CAPE (Gettelman et al., 2002; DeMott and Randall, 2004; Riemann-Campe et al., 2008; Brooks, 2013). Hence, CAPE can also be used as a potential indicator of climate change (Murugavel et al., 2012). Thus, the behaviour of convection can be partially addressed by understanding the changes in CAPE.

Variability in CAPE can also affect the temperature field in the upper troposphere. Gaffen et al. (1991) discussed the dynamical link between lower tropospheric CAPE and variations in the temperature in the upper troposphere. Dhaka et al. (2010) studied the relationship between seasonal, annual, and large-scale variations in CAPE and the solar cycle on the temperature at 100 hPa pressure levels using daily radiosonde data for the period 1980–2006 over the Indian region. They showed that the increase in CAPE was associated with the decrease in temperature at 100-hPa pressure level on all time scales.

The convective schemes in general circulation models use CAPE as a variable for calculating convective heating (e.g., Arakawa and Schubert, 1974; Moncrieff and Miller, 1976; Washington and Parkinson, 2005). Many of the cumulus parameterization schemes make use of CAPE in constructing closures (Donner and Phillips, 2003). The diurnal variation of CAPE is of primary importance for understanding the sensitivity of convection schemes in the model to produce the diurnal cycle of precipitation (Lee et al., 2007). The reliability of model-simulated temporal and spatial variations in CAPE is an important indicator of model performance, particularly in the tropics (Gettelman et al.,





2002). Also, the seasonal and diurnal changes in CAPE are important for models to provide validation

of their capacity to simulate future changes in the tropical climate. The above studies conclude that the

estimation of CAPE are imperative, not only for assessing the conditional instability of the atmosphere

and for the convective parameterization, but also for the studies related to climatic change.

Most of the earlier studies on CAPE are based on radiosonde observations. Globally, the radiosonde is

launched twice a day, 0000 and 1200 UTC. This limits the studies on CAPE at diurnal scales. It is also

to be noted that the radiosonde observations are limited to land, and are very sparse over the oceans.

Reanalysis datasets fill these gaps; however, their spatial resolution is poor and most of the time the

data accuracies do not match with the standard techniques. The satellite observations are the only

solutions to have regular observations of CAPE with high spatial resolution across the globe. With the

availability of satellite measurements, several studies were carried out on CAPE. Narendra Babu et al.

(2010) studied the seasonal and diurnal variations in CAPE over land and oceanic regions using one

year of observations from the FORMOSAT mission 3/Constellation Observing System for

Meteorology, Ionosphere, and Climate (COSMIC/FORMOSAT-3) Global Positioning System (GPS)

Radio Occultation (GPS RO) measurements. In order to study the diurnal variation of stability indices,

they integrated the data over a season as the occultations were sparse and hence not adequate to study

on daily scale. This can be overcome by the use of geostationary satellites. These geostationary satellite

measurements provide near-continuous monitoring of instability in the atmosphere with better spatial

coverage, which is helpful in nowcasting of convection (Koenig and de Coning, 2009). Siewert et al.

(2010) discussed the advantages of the METEOSAT Second Generation (MSG) system in deriving the

instability indices and to predict the convection initiation over the Central Europe and South Africa.

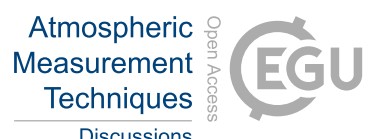

Using the MSG satellite measurements, de Coning et al. (2011) derived a new convection indicator, the

combined instability index which can calculate the probability of convection over the South Africa.

They showed that the combined instability index can predict the convection better than the individual

instability indices like K-index, total totals index etc. Jewett and Mecikalski (2010) developed an

algorithm to derive convective momentum fluxes from the Geostationary Operational Environmental

Satellite (GOES) measurements. The advantage of this algorithm is that it can be used in any convective

environment. Botes et al. (2012) investigated the performance of the Atmospheric Infrared Sounder

(AIRS) soundings data with the collocated radiosonde observations. They showed that the AIRS

measurements underestimate instability due to dry bias at the surface.

Recently, the Indian Space Research Organization (ISRO) launched the Indian National Satellite

System (INSAT-3D), which is a geostationary satellite that provides the profile of temperature and

relative humidity with high temporal and spatial resolution. The purpose of the INSAT-3D

measurement is to enhance the understanding of the atmospheric processes and also to monitor land and

oceans in order to accurately forecast weather and to manage disasters. Several researchers evaluated

the temperature and relative humidity measurements from the INSAT-3D. Mitra et al. (2015) evaluated

the INSAT-3D temperature and moisture retrievals up to 100 hPa with GPS sonde observations for the

period January-May 2014. They observed that the INSAT-3D measurements compare better with GPS

sonde observations at middle levels (from 900 hPa to 500 hPa). The assessment of the quality of

temperature and water vapour obtained from the INSAT-3D with in-situ, satellite, and reanalysis

datasets by Ratnam et al. (2016) revealed that the INSAT-3D measurements agree well with the other

satellite measurements and reanalysis datasets below 25$^o$N. The temperature difference was 0.5K with a

standard deviation of about 1K, and for humidity, a dry bias (20-30%) was observed between INSAT-3D and other satellite measurements and reanalysis data. Hence, these satellite measurements also suffer from some inherent shortcomings and have biases and random errors. Therefore, it is essential to evaluate the satellite products with conventional measurements to quantify the direct usability of these products.

The objective of the present study is to quantitatively evaluate the accuracy of CAPE estimated from the INSAT-3D measurements with radiosonde measurements over the Indian region. Here, an attempt is made first to validate the estimated CAPE from the INSAT-3D measurements with that of radiosonde measurements over different stations in India. In general, there are many profiles that do not reach the ground level in the INSAT-3D measurements. Hence, different statistical indices are calculated to assess the detectability of INSAT-3D derived CAPE over these regions. Secondly, the diurnal variation of CAPE is studied at different regions in India. Finally, the seasonal mean CAPE is estimated over the Indian region. The paper is organized as follows. Section 2 provides the details of the data sets used. Section 3 discusses the methodology adopted to estimate the CAPE. Section 4 provides the results and discussion. Finally, the summary of the present study is provided in section 5.

## 2. Database

### 2.1. *INSAT-3D*

In the present study, three years (01 April 2014 - 31 March 2017) of measurements obtained from the INSAT-3D are used to estimate CAPE over the Indian region and assess the estimation against radiosonde measurements. The INSAT is a series of multipurpose geostationary satellites launched by



the ISRO, India. The INSAT-3D, which is considered to be the advanced version of all the other INSAT series satellites, is a multipurpose geosynchronous spacecraft with main meteorological payloads (imager and sounder) launched on 26 July 2013. The main objective of the mission is to monitor the

earth and ocean continuously and also provide data dissemination capabilities. The INSAT-3D also provides an operational, environmental and storm warning system to protect life and property.

The INSAT-3D spacecraft carries two meteorological payloads: (i) Imager (optical radiometer) provides high-resolution images of mesoscale phenomena in the visible and infrared (IR) spectral bands (0.55 to 12.5 μm) and (ii) Sounder has one visible and 18 IR (7 in long-wave IR, 5 in mid-IR, and 6 in short-

wave IR) channels. The sounder measures the irradiance and provide profiles of temperature, water vapour and integrated ozone over the Indian landmass and surrounding ocean every hour and over the whole of the Indian Ocean every 6 hours with a spatial resolution of $0.1^o$. This is the simplest scanning mode in which the soundings are available every hour over larger land and ocean region. For the present study, temperature and water vapour data collected from the INSAT-3D sounder are interpolated to

$0.25^o$ spatial resolution and are used to estimate the CAPE over the Indian region.

### 2.2. *Radiosonde*

Upper air radiosonde profiles are downloaded from the University of Wyoming website (http://weather.uwyo.edu/upperair/sounding.html). Radiosonde data are usually available at 0000 and 1200 UTC regularly to monitor the thermodynamic state of the atmosphere by the National Weather

Service. For the present study, the data collected for 20 stations (black dots in Figure 1) for the period from 01 April 2014 to 31 March 2017 are used to assess the INSAT-3D estimated CAPE. The values of CAPE reported in this paper are taken directly from the data provided by the University of Wyoming.

## 2.3. *ERA-Interim Reanalysis*

The reanalysis dataset used in this study are from the ERA-Interim project (Dee et al., 2011). The ERA-

Interim is the latest global atmospheric reanalysis produced by the European Centre for Medium-Range

Weather Forecasts (ECMWF), which envisaged preparing a future reanalysis project that will span the

entire twentieth century. The ERA-Interim data are available in near-real time from 1 January 1979

onwards. The ERA-Interim generates gridded data, including a large variety of surface parameters that

describe the weather as well as land surface and ocean conditions at the 3-hourly and 6-hourly interval.

The present study utilizes the CAPE derived from the ERA-Interim dataset. The data are extracted over

the Indian region at 0000, 0300, 0600, 0900, 1200, 1500, 1800 and 2100 UTC for each day between 01

April 2014 and 31 March 2017. The spatial resolution of the data utilized is $0.75^{o} \times 0.75^{o}$.

## 3. Estimation of CAPE

To calculate the CAPE over the Indian region, the vertical profiles of pressure, temperature, and water

vapour are taken from the INSAT-3D measurements. For many years, a debate has existed in the

literature regarding the most meaningful way to calculate CAPE and how to interpret the result. The

most important methods of calculating CAPE is (1) pseudoadiabatic CAPE in which CAPE is estimated

after assuming that all the condensate has fallen out of the air parcel and (2) reversible CAPE in which

it is assumed that the condensate remains within the parcel. In the present study, the pseudoadiabatic

algorithm is used to calculate CAPE over the Indian region. Further, CAPE is very sensitive to near-

surface temperature and humidity (Gartzke et al., 2017), which are known to vary spatially. However,

Ratnam et al. (2016) found that the variability of the atmospheric state within the INSAT-3D footprint





sampled by the radiosonde had little bias over the Indian region. They also reported that the difference

in temperature between INSAT-3D and ERA-Interim reanalysis datasets lies within 1K and a dry bias

of 5–10% was found in the lower and mid-troposphere relative humidity when compared with the ERA-

Interim reanalysis datasets. In this scenario, it is assumed that the spatial sampling mismatch may not

affect much in the calculation of CAPE and hence are neglected in the present study. In addition, only

cases with radiosonde profiles having CAPE greater than 0 J kg$^{-1}$ are included in the analysis. This

threshold is used to eliminate the large number of zero CAPE values.

The integration of the buoyancy of the air parcel from the level of free convection (LFC) to equilibrium

level (EL) gives the measure of CAPE.

$$CAPE = \int_{LFC}^{EL} \frac{g(T_{vp} - T_{ve})}{T_{ve}} dZ$$

Where, $T_{vp}$ is the virtual temperature of the air parcel and $T_{ve}$ is the virtual temperature of the

environment, g is the acceleration due to gravity. The LFC is situated above the lifting condensation

level (LCL) and at that level, the parcel temperature is greater than the environmental temperature. This

is calculated by lifting the air parcel moist adiabatically. The EL is situated above the LFC and at this

level, the parcel temperature is less than or equal to the environment temperature. At EL, the air parcel

attains stability and the convection stops. Under stable environmental conditions, the LFC and EL will

not be present. The procedure to estimate CAPE is similar as discussed in Uma and Das (2017).

## 4. Results and Discussions



Three years of data collected from the INSAT-3D measurements are utilized to estimate CAPE over the Indian region. These estimates are compared with the radiosonde derived CAPE at 0000 and 1200 UTC along with the ERA-Interim reanalysis CAPE data.

### 4.1. *Comparison of INSAT-3D and ERA-Interim estimated CAPE with radiosonde measurements*

In this work, 20 stations are selected for which the radiosonde profiles are available during the study period (location shown in Fig. 1). Fig. 2 shows the correlation coefficient along with the number of data points used in the analysis for the comparison between the estimated CAPE from the INSAT-3D measurements, ERA-Interim CAPE, and radiosonde derived CAPE. From this figure, it is noticed that the INSAT-3D estimated CAPE shows the better correlation with radiosonde CAPE compared to the ERA-Interim CAPE. In general, the coastal stations show a higher correlation coefficient than that of the other stations for the INSAT-3D estimated CAPE. All the coastal stations show a correlation higher than 0.6 except Trivandrum. The correlation coefficient is as high as 0.84 for Chennai among all the stations. The correlation values are lower for the stations located near the foothills of the Himalayas and north-east (NE) regions of India. A weak correlation of 0.31 is found for Delhi. The ERA-Interim estimated CAPE shows less correlation coefficient for all the stations except Agarthala compared to the INSAT-3D estimated CAPE. Even in the ERA-Interim CAPE, the coastal stations show better correlation compared to other stations. The ERA-Interim CAPE shows a higher correlation for Agarthala and the correlation is minimum for Port Blair. This result suggests that the INSAT-3D CAPE measurements agree well with the radiosonde measurements. It is worth to note that the comparison shows better correlation for the stations where the number of data is higher.



To elucidate the consistency of INSAT-3D estimated CAPE against the ERA-Interim CAPE, the bias in

the measurements of INSAT-3D and ERA-Interim with the radiosonde is presented in Fig. 3. The bias

evaluates the size of the difference between the two datasets. The positive (negative) value of bias

indicates the overestimation (underestimation) of the satellite/reanalysis measurements. The INSAT-3D

estimated CAPE shows small and positive bias for most of the stations considered in the study. Among

all the stations, Hyderabad, Gorakhpur, and Delhi show higher bias, whereas Mumbai shows minimum

bias. All the stations show positive bias except Ahmedabad, Dibrughar, Delhi, Gorakhpur, and Port

Blair where the bias is negative and Mumbai shows small negative bias. Whereas, the ERA-Interim

CAPE shows negative bias for all the stations indicating the underestimation of CAPE compared to

radiosonde measurements. Further, the bias in the estimation of CAPE in the ERA-Interim data is

higher for most of the stations. Among all the stations, Gorakhpur shows higher negative bias. From

this, it is clear that the INSAT-3D (ERA-Interim) overestimated (underestimated) the CAPE compared

to radiosonde measurements. From the above discussion, it can be concluded that the INSAT-3D

provides better estimates over coastal regions compared to other regions and also a better comparison

with the radiosonde measurements.

**4.2.** *Statistical indices in the comparison of INSAT-3D estimated CAPE*

Furthermore, to examine the capability of detection of the INSAT-3D sounder, the probability of

detection (POD), false alarm ratio (FAR), critical success index (CSI), and accuracy (ACC) are

computed on the basis of a contingency table (Table 1). A threshold value of zero is considered for

CAPE to be estimated by the satellite measurements. The POD is a measure of the CAPE successfully

identified by the satellite product, and FAR gives a proportional measure of the satellite product's



tendency to estimate CAPE where none is observed i.e., it gives the CAPE estimates that are incorrectly

detected. CSI represents how well the estimated CAPE events correspond to the observed CAPE events.

ACC measures the fraction of the correctness in the CAPE estimates. For a perfect satellite-based

estimate, the values of POD, FAR, CSI, and ACC should be 1, 0, 1, and 1 respectively. These statistical

indices can be calculated as:

$$POD = \frac{a}{a + c}$$


$$FAR = \frac{b}{a + b}$$

$$CSI = \frac{a}{a + b + c}$$

$$ACC = \frac{a + d}{a + b + c + d}$$

Figure 4(a)-(d) shows the POD, FAR, CSI, and ACC calculated for the INSAT-3D estimated CAPE.

Among all the stations, Mumbai shows higher POD whereas Hyderabad shows lower POD. This

indicates that the INSAT-3D is unable to detect CAPE over Hyderabad. This may be due to less

availability of data and the inability to catch the short-lived convective storms frequently observed over

this region. All the coastal stations show the higher POD, CSI, and ACC. Cochin shows the highest CSI





and ACC, whereas Agarthala and Kolkata have highest FAR. This indicates that the INSAT-3D product

performs reasonably well with the higher POD, CSI, and ACC and lower FAR.

### 4.3. *Distribution of CAPE over India*

A summary of the frequency distribution of CAPE computed from the INSAT-3D, ERA and radiosonde

measurements for the 20 stations are shown in Figure 5. The distribution of CAPE shows the higher

occurrence for lower values. Thus, the CAPE distribution is shifted to lower values in all the three

measurements. The INSAT-3D estimated CAPE and ERA-Interim CAPE shows a similar kind of

distribution compared to radiosonde CAPE. The distribution of the INSAT-3D estimated CAPE is

higher (lower) in the range ~300-1200 (1200-3000) J kg$^{-1}$ compared to the radiosonde measurements.

The INSAT-3D estimated CAPE matches with the radiosonde CAPE above 3000 J kg$^{-1}$. The ERA-

Interim CAPE distribution shows higher values below 1200 J kg$^{-1}$. However, the ERA-Interim

distribution becomes negligible above 2000 J kg$^{-1}$. This also shows that for the higher values of CAPE,

the ERA-Interim underestimates the observations. This may be due to the fact that the spatial resolution

of the reanalysis data are coarse compared to the observations. The figure shows that among the two

distributions, the INSAT-3D distribution agrees well with the radiosonde distribution.

### 4.4. *Seasonal variation of CAPE*

In the present study, we have divided the period into four seasons viz., winter (December to February),

pre-monsoon (March to May), monsoon (June to September), and post-monsoon (October and

November). Fig. 6(a)-(d) shows the seasonal mean CAPE estimated from the INSAT-3D measurements

during the period from April-2014 to March-2017 over the Indian region. During the winter season (Fig.

6a), the mean CAPE is below 500 J kg$^{-1}$ over the land regions. The mean CAPE is relatively higher over



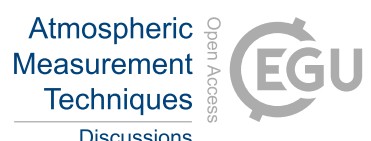

the west coast and Arabian Sea (AS) and the parts of northern Bay of Bengal (BoB). This relatively

high CAPE over oceans and the western coastal regions may be due to the occurrence of depression and

cyclone during the month of December over the oceanic regions. In the pre-monsoon season (Fig. 6b),

the higher values of CAPE (above 2000 J kg$^{-1}$) are over the AS, BoB and Central India (CI). The pre-

monsoon depressions are regular during this time period over the surrounding oceanic regions of India.

This sustains for few days which results in higher CAPE over these regions. Higher values of CAPE are

the causing factors for frequent thunderstorms and deep convection over the Northern and Central India.

The Western Ghats has higher CAPE which may be due to the orographic induced deep convection.

CAPE is lower over the Southern Peninsular India. Further, the CI, north India (NI) and foothills of

Himalayas also exhibits higher CAPE compared to other land regions. The north-western parts of India

(Gujarat) show higher CAPE among other land regions. During the monsoon season (Fig. 6c), the east

coast of India, AS, BoB, and foothills of Himalayas shows relatively higher CAPE than the other

regions. During the post-monsoon period (Fig. 6d), the northern parts of AS and BoB, east coast of

India and west coast of India shows higher values of CAPE and the southern peninsula and eastern

regions shows lower CAPE.

Further, the spatial distribution of CAPE estimated from the INSAT-3D is studied for different CAPE

ranges. The spatial distribution of CAPE provides information on the distribution of extreme weather

events over the study region. The estimated CAPE is divided into four categories: weak instability

(<500 J kg$^{-1}$), moderate instability (501-1500 J kg$^{-1}$), strong instability (1501-3000 J kg$^{-1}$), and extreme

instability (>3000 J kg$^{-1}$). The normalized anomaly distribution of CAPE in the four instability

conditions during different seasons is provided over the Indian region as shown in Fig. 7. The spatial



distribution of CAPE during the weak, moderate, strong, and extreme instability conditions is shown in

Fig. 7(a)-(d), 7(e)-(h), 7(i)-(l), and 7(m)-(p) respectively. The top panel is for the winter, second from

the top is for the pre-monsoon, third from the top is for the monsoon and bottom panel is for the post-

monsoon season. Here, the negative (positive) anomaly indicates the increase (decrease) in CAPE.

During the winter, the response of weak instability (Fig 7a) is very less over the Indian subcontinent as

well as the surrounding oceanic regions. However, the response of winter towards strong and extreme

instability is observed over the BoB, Southeast AS, and some parts of the NI. The higher response may

be due to the cyclones and depressions that occur over the oceanic regions during December. Over the

NI, strong westerlies are observed during the winter, which may result in a dry convection with higher

CAPE.

The response during pre-monsoon is observed to be high during strong and extreme instability. The pre-

monsoon season is considered to be summer over the Indian subcontinent and it is the favorable season

for thunderstorms and deep convection. This could be easily observed in the figure that during extreme

instability the whole Indian region is observed to have very high values of CAPE. During the monsoon,

the response to weak and moderate instability is more compared to strong and extreme instability. It is

observed that the Western Ghats and the surrounding oceanic regions have more frequency compared to

the other regions. The Western Ghats is generally dominated by the shallow clouds (e.g., Das et al.,

2017; Utsav et al., 2017), which results its high response with weak instability as deep convection does

not predominantly occur over this region. The monsoon trough region extending from heat low in

Pakistan to head BoB respond to strong instability. This trough (core low-pressure region) occurs during

the monsoon, which results in heavy rainfall over the Indian subcontinent. Compared to the pre-



monsoon, there is little response in strong instability conditions over the oceanic regions. This results in

fewer occurrences of deep convection and cyclones (inhibited because of the presence of strong wind

shear at 500 hPa) during the monsoon season. In the post-monsoon, the response is much similar to the

pre-monsoon during the strong and extreme instability conditions. In the post-monsoon, the wind flow

over the Indian region is northeasterly which results in more convection over the northwestern and the

southern peninsular Indian region. Usually, in October-November months, the deep depression occurs

over the Bay of Bengal due to which the frequency of higher CAPE is observed. In general, the weak

instability is predominant during the winter season. The moderate instability shows higher occurrence

during the post-monsoon compared to other seasons. The strong instability condition is more during the

monsoon period whereas the occurrence of extreme instability is higher during the pre-monsoon

months.

**4.5.** *Diurnal variation of CAPE*

Fig. 8 shows the hourly mean CAPE averaged for three years (2014-2017). A strong diurnal variation in

the mean CAPE is observed over different parts of India. A clear land-sea contrast is also observed in

the mean CAPE. The CAPE starts building up in the morning (0530 LT=00UTC+0530) over the oceans

with land having low values of CAPE. The mean CAPE reaches its maximum at ~1200 LT over the

oceans. However, the CAPE starts increasing after 0900 LT over the land and reaches the maximum in

the afternoon (between 1300 and 1400 LT) and decreases again thereafter. The land-sea contrast in the

mean CAPE has disappeared in the evening. Again in the evening, the CAPE increases and a secondary

maximum is observed in the midnight over the oceans. Over the tropics, the Indian region is one of the

active convective regions. The deep convective clouds form during the daytime over the Indian sub-





continent due to solar insolation, which increases the lower tropospheric temperature resulting in convective instability. Uma and Das (2016) have found the lower tropospheric humidity maximum in

the afternoon and minimum in the evening hours. The surrounding oceanic regions found to peak in the late evening and midnight. This indicates that the convection peaks in the late afternoon over the land region and evening to midnight over the oceans. These results are consistent with the findings of Dutta and Rao (2001) and Dutta and Kesarkar (2004). These studies revealed that the maximum value of CAPE is observed during nighttime over the BoB and east coast of India. However, this secondary

maximum is not observed over the land areas. In contrast, another maximum is observed around 1700 LT over the east coast, west coast and north-west regions of India.

To observe the diurnal variation of CAPE over different parts of India, the study region is divided (latitude-wise for uniformity) into six sub-regions. These regions are: the Arabian Sea (AS; 8-20$^o$N, 65-72$^o$E), Bay of Bengal (BoB; 8-20$^o$N, 80-90$^o$E), South Peninsular India (SP; 8-20$^o$N, 72-80$^o$E), Central

India (CI; 20-25$^o$N, 73-82$^o$E), North India (NI; 25-35$^o$N, 73-80$^o$E), and Northeast India (NE; 24-29$^o$N, 90-96$^o$E). Fig. 9 shows the diurnal variation of mean CAPE over these sub-regions during four seasons. From this figure, one can observe a bimodal distribution in the mean CAPE over the AS and BoB during the pre-monsoon and monsoon periods (Fig. 9a and b). The primary maximum is observed in the nighttime and the secondary maximum is observed in the afternoon time over these regions. Uma and

Das (2016) have also observed bimodal distribution in the relative humidity over the Bay of Bengal and the Indian Ocean. They found maximum distribution at 1200 and 1500 LT, which is almost similar to that observed in the present study. During Bay of Bengal Monsoon Experiment (BOBMEX) 1999, Dutta and Kesarkar (2004) observed that the CAPE is maximum in the nighttime than in the daytime.





Further, the nighttime maximum in the mean CAPE is observed around ~0000 LT over BoB, and AS

during the pre-monsoon and monsoon periods. The bimodal distribution in mean CAPE is also observed

over the AS and BoB during the post-monsoon season. However, the secondary maximum is not much

prominent as observed in the pre-monsoon and monsoon periods. The mean CAPE peaks in the

afternoon hours over the AS and BoB during the winter season.

The SP region (Fig. 9c) also shows the similar behavior as that of oceans. The mean CAPE over the SP

region maximizes in the afternoon hours. The secondary maximum is also observed in the late night

(0000 LT). The mean CAPE in the CI region (Fig. 9d) shows a bimodal distribution during the winter,

monsoon and post-monsoon seasons. The mean CAPE is at its maximum in the afternoon time. The

other maximum is during the nighttime. A noticeable difference is observed in the mean CAPE during

the pre-monsoon season. During this period, the CAPE is nearly the same throughout the day. However,

this CAPE decreases a little and becomes minimum in the early morning over the CI region. The mean

CAPE in the NI region (Fig. 9e) shows a similar diurnal variation as observed over the CI region. The

mean CAPE shows little variation during the pre-monsoon months. The other three seasons show

bimodal distribution as observed in other regions. However, the magnitude of the mean CAPE is higher

in the NI region compared to the CI region. Further, the difference in mean CAPE between primary and

secondary maximum is relatively small compared to the other regions. The NE region (Fig. 9f) also

shows a bimodal distribution in mean CAPE with the primary maximum in the afternoon hours and the

secondary maximum in the late-night during the winter, monsoon and post-monsoon seasons. In

addition to these two maxima, a third maximum is also observed over the NE region during the pre-

monsoon season. The third maximum is observed at ~0900 LT over this region. The third maximum



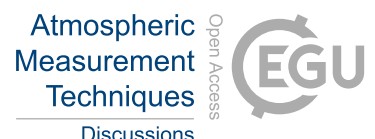

observed in the mean CAPE may be due to the occurrence of pre-monsoon thunderstorms known as

Norwesters during the morning over this region.

The statistical analysis of CAPE is also attempted over these six regions to understand the variability

between the different regions during different seasons and it is shown in Fig. 10(a)-(d) for the winter,

pre-monsoon, monsoon, and post-monsoon respectively. The mean, standard deviation along with

maximum/minimum with 25, 50 and 70 % occurrence are provided in the figure. During the winter, the

mean CAPE is higher (~1600 J kg$^{-1}$) over the AS compared to that of the BoB (~1000 J kg$^{-1}$). Over the

land regions, the mean CAPE is found to be less than ~1000 J kg$^{-1}$. The oceanic regions are found to

have higher CAPE during the winter compared to that of the continent. The NI and NE have smaller

CAPE during the winter. The winter is extremely dry over the Indian region except for the surrounding

oceanic regions and south peninsular India, where we observe cyclones/depressions during December,

which brings more moisture and heavy rainfall as discussed earlier. During the pre-monsoon, the mean

CAPE is higher than ~2000 J kg$^{-1}$ over all the regions concerned. The pre-monsoon depressions,

thunderstorms, deep convection and Norwesters contribute to very high CAPE over India and

surrounding oceanic regions. During the monsoon, the mean CAPE is less than ~1500 J kg$^{-1}$ over all the

regions except the NI where it is about ~2000 J kg$^{-1}$. The shallow convection dominates during the

monsoon season rather than deep convection, which results in lesser CAPE compared to that of the pre-

monsoon. During the post-monsoon, the mean value of CAPE is about ~1200 J kg$^{-1}$ except AS where it

is about ~1800 J kg$^{-1}$  and BoB about ~1500 J kg$^{-1}$. The northeast monsoon dominates the Indian region

during the post-monsoon which results in deep convection resulting in relatively higher values of

390 CAPE. Overall, over the Indian and the surrounding oceanic regions, the maximum CAPE is found in

the pre-monsoon followed by the post-monsoon, monsoon and winter.

## 5. Summary

The extreme weather events such as thunderstorms and tropical cyclones cause severe damage to life

395 and property, especially in the tropical regions. Convective available potential energy (CAPE) is a

measure of the amount of energy available for convection in the atmosphere. Hence, CAPE can be used

as a measure for the occurrence of these severe weather conditions. In the present study, we made an

attempt for the first time to estimate CAPE from the INSAT-3D measurements and evaluate

comprehensively over the Indian region. For the evaluation, 20 stations are selected in different parts of

400 India and these estimates are evaluated against the radiosonde measurements collected from the

Wyoming University along with the ERA-Interim data. The station wise comparison shows that the

INSAT-3D estimates match well with higher correlation coefficient and lower bias with the radiosonde

measurements. The correlation coefficient between INSAT-3D and radiosonde CAPE is higher than that

between ERA-Interim and radiosonde. Further, the INSAT-3D derived CAPE overestimate the CAPE

405 compared to radiosonde measurements, whereas the ERA-Interim estimates underestimate the

radiosonde CAPE. The categorical statistics shows that the INSAT-3D can better represent the

radiosonde measured CAPE. The distribution of CAPE collected from all the 20 stations shows that the

CAPE distribution is shifted to lower values in all the three measurements. The INSAT-3D and ERA-

Interim estimated CAPE is higher than the radiosonde measurements in the lower ranges. The INSAT-

410 3D estimated CAPE matches well with the radiosonde measurements above ~3000 J kg$^{-1}$.

The spatial and temporal distribution of CAPE reveals several interesting features. The weak instability is predominant during the winter season, the moderate instability is higher during the post-monsoon, the strong instability is more during the monsoon period and the extreme instability is higher during the pre-monsoon months. The diurnal variation in mean CAPE shows a bimodal distribution with the primary peak around mid-night and secondary peak in the afternoon times for most of the regions and in different seasons. The seasonal mean CAPE shows that the land areas show lower CAPE during the winter, whereas the oceans show the highest CAPE during the pre-monsoon season. The higher values of CAPE over the oceanic regions may be due to the higher sea surface temperature and higher occurrence of the tropical cyclones during the pre-monsoon season. Further, the north-western parts of India (Gujarat) show higher CAPE among other land regions. Overall, the INSAT-3D estimated CAPE is in close agreement with the radiosonde derived CAPE. As the INSAT-3D provides high temporal and spatial resolution data, hence it can be used for now-casting and severe weather warnings in the numerical prediction models.

**Acknowledgement**

The authors would like to acknowledge Meteorological and Oceanographic Satellite Data Archival Centre (MOSDAC) of Space Application Centre (ISRO) for supplying the INSAT-3D data. The authors are grateful to the Department of Atmospheric Science, University of Wyoming for access to their radiosonde data archive. The authors acknowledge the NOAA and the ERA-Interim team for providing their data. All the dataset are freely available and can be downloaded from their respective archive. The



authors would like to sincerely thank the Editor for his insightful comments that improved the quality of

the manuscript.



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



**Table Captions:**

**Table 1:** Contingency table for the comparison of INSAT-3D estimated CAPE with radiosonde measured CAPE. The CAPE threshold is assumed as 0 J kg$^{-1}$.

**Figure Captions:**

**Fig 1:** The geographical map of India representing the 20 stations considered for the present study. Stations are Agarthala (AGR; 23.88˚N, 91.25˚E); Ahmedabad (AHB; 23.06˚N, 72.63˚E); Amini Divi (AMD; 11.12˚N, 72.73˚E); Bhuvaneswar (BHU; 20.25˚N, 85.83˚E); Chennai (CHE; 13.00˚N, 80.18˚E); Cochin (COC; 9.95˚N; 76.26˚E); Delhi (DEL; 28.58˚N, 77.20˚E); Dibrugarh

(DIB; 27.48˚N, 95.01˚E); Gorakhpur (GRK; 26.75˚N, 83.36˚E); Guwahati (GUW; 26.10˚N; 91.58˚E); Hyderabad (HYD; 17.45˚N, 78.46˚E); Karaikal (KAR; 10.91˚N, 79.83˚E); Kolkata (KOL; 22.65˚N, 88.45˚E); Machilipatnam (MAP; 16.20˚N, 81.15˚E); Mangalore (MAN; 12.95˚N, 74.83˚E); Minicoy (MIN; 8.30˚N, 73.15˚E); Mumbai (MUM; 19.11˚N, 72.85˚E); Port Blair (PB; 11.66˚N, 92.71˚E); Trivandrum (TVM; 8.48˚N, 76.95˚E); Visakhapatnam (VSP;

17.70˚N, 83.30˚E).

**Fig 2: (a)** Correlation Coefficient in the comparison of INSAT-3D and ERA CAPE with radiosonde derived CAPE for 20 stations in India for the Period April, 2014 to March, 2017. **(b)** Number of data points in the comparison of INSAT-3D-CAPE, ERA-CAPE with radiosonde CAPE.

**Fig 3:** Bias in the comparison between INSAT-3D-CAPE and ERA reanalysis CAPE with radiosonde

for different Stations in India for the Period April, 2014 – March, 2017.





**Fig 4:** Statistical indices POD, FAR, CSI and ACC in the comparison between the INSAT-3D-CAPE

with radiosonde CAPE for 20 Stations in India.

**Fig 5:** Distribution (%) of INSAT-3D, radiosonde and ERA reanalysis CAPE (J kg$^{-1}$) for all the stations

considered in the study.

**Fig. 6:** The seasonal mean distribution of CAPE (J kg$^{-1}$) during (a) winter (b) pre-monsoon (c) monsoon

and (d) post-monsoon from INSAT-3D data over Indian region.

**Fig. 7:** Frequency distribution (%) of CAPE in (a) weak (b) moderate (c) strong (d) extreme instability

during winter season. (e)-(h): same as (a)-(d) except for pre-monsoon season. (i)-(l): same as

(a)-(d) but for monsoon season and (m)-(p): same as (a)-(d) except for post-monsoon season.

**Fig 8:** Hourly mean distribution of the INSAT-3D CAPE (J kg$^{-1}$) during the period from 01 April 2014

to 31 March 2017 over the Indian region.

**Fig 9:** Diurnal variation of CAPE (J kg$^{-1}$) over (a) AS (b) BoB (c) SP (d) CI (e) NI (f) NE region of

India during four monsoon seasons.

**Fig 10:** Box-plot analysis of distribution of CAPE (J kg$^{-1}$) during (a) winter (b) pre-monsoon (c)

monsoon and (d) post-monsoon from INSAT-3D data over the Indian region.



**Table 1:** Contingency table for the comparison of INSAT-3D estimated CAPE with radiosonde measured CAPE. The CAPE threshold is assumed as 0 J kg$^{-1}$.

|  | Radiosonde ≥Threshold | Radiosonde < Threshold |
|---|---|---|
| Satellite ≥Threshold | Hits (a) | False alarms (b) |
| Satellite <Threshold | Misses (c) | Correct negatives (d) |





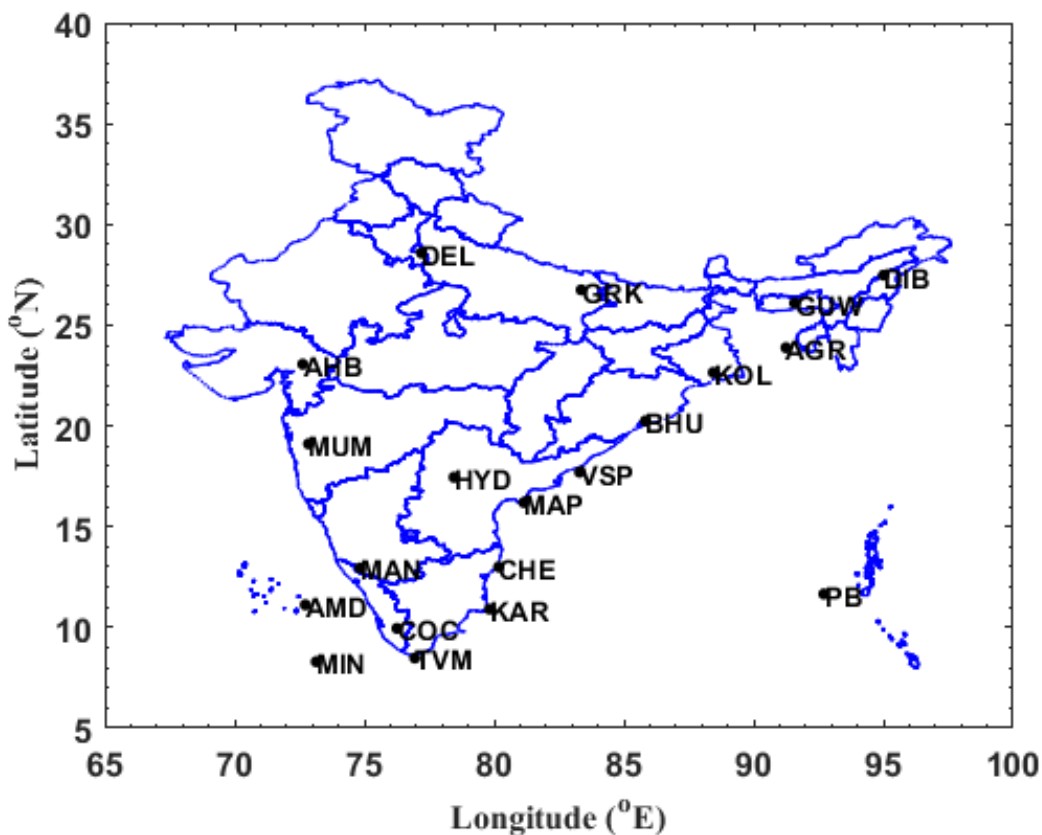


**Fig 1:** The geographical map of India representing the 20 stations considered for the present study.

Stations are Agarthala (AGR; 23.88˚N, 91.25˚E); Ahmedabad (AHB; 23.06˚N, 72.63˚E); Amini Divi

(AMD; 11.12˚N, 72.73˚E); Bhuvaneswar (BHU; 20.25˚N, 85.83˚E); Chennai (CHE; 13.00˚N, 80.18˚E);

Cochin (COC; 9.95˚N; 76.26˚E); Delhi (DEL; 28.58˚N, 77.20˚E); Dibrugarh (DIB; 27.48˚N, 95.01˚E);

Gorakhpur (GRK; 26.75˚N, 83.36˚E); Guwahati (GUW; 26.10˚N; 91.58˚E);    Hyderabad (HYD;

17.45˚N,  78.46˚E); Karaikal (KAR;  10.91˚N,  79.83˚E); Kolkata  (KOL;  22.65˚N,  88.45˚E);

Machilipatnam (MAP; 16.20˚N, 81.15˚E); Mangalore (MAN; 12.95˚N, 74.83˚E); Minicoy (MIN;

8.30˚N, 73.15˚E); Mumbai (MUM; 19.11˚N, 72.85˚E); Port Blair (PB; 11.66˚N, 92.71˚E); Trivandrum

(TVM; 8.48˚N, 76.95˚E); Visakhapatnam (VSP; 17.70˚N,  83.30˚E).



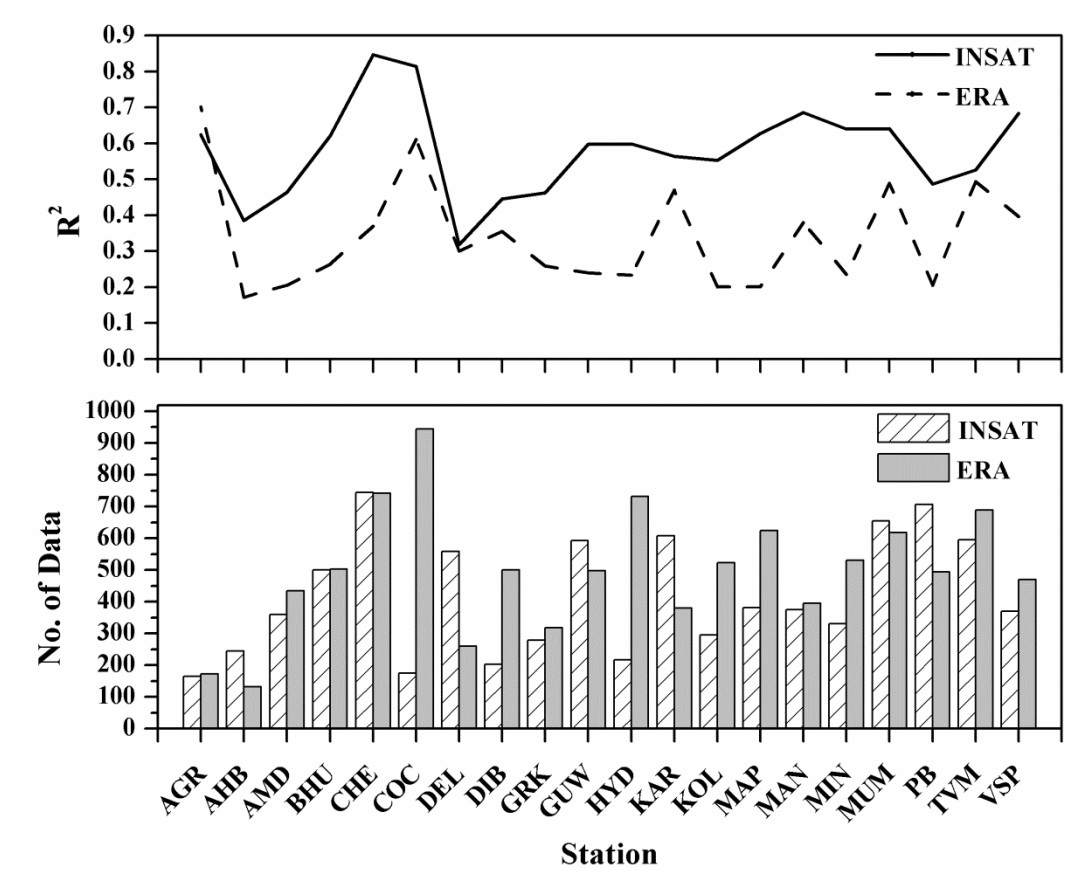


**Fig 2: (a)** Correlation Coefficient in the comparison of INSAT-3D and ERA CAPE with radiosonde

derived CAPE for 20 stations in India for the Period April, 2014 to March, 2017. **(b)** Number

of data points in the comparison of INSAT-3D-CAPE, ERA-CAPE with radiosonde CAPE.





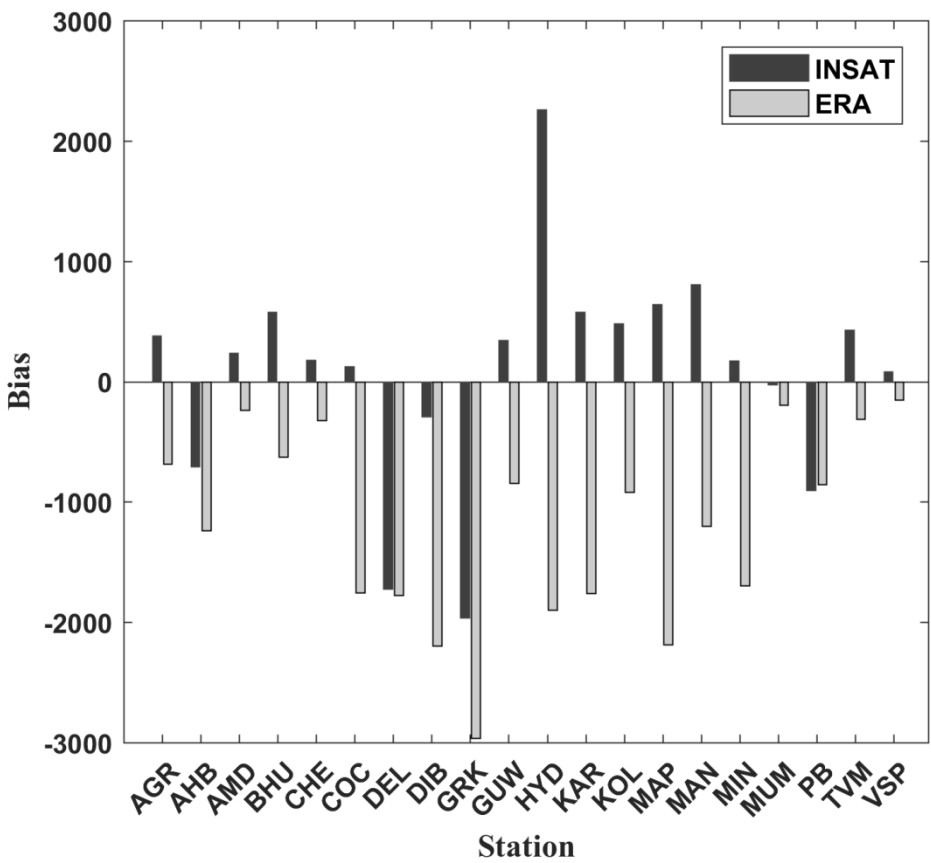

**Fig 3:** Bias in the comparison between INSAT-3D-CAPE and ERA reanalysis CAPE with radiosonde

for different Stations in India for the Period April, 2014 – March, 2017.





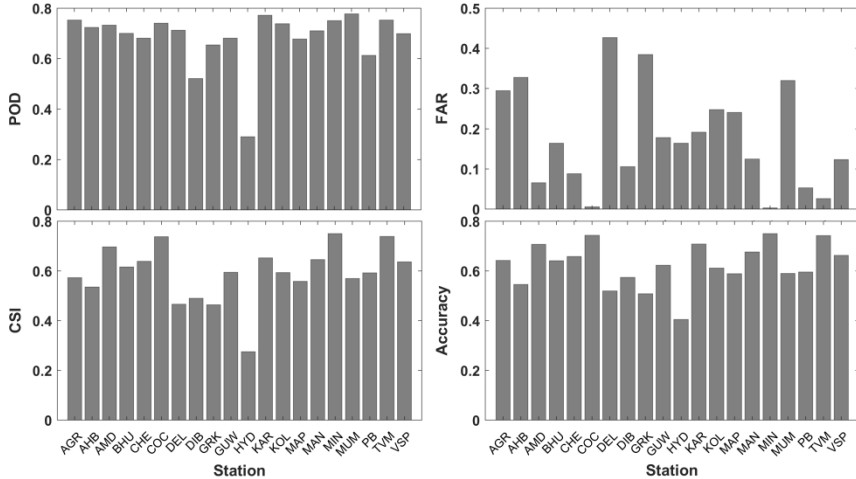

**Fig 4:** Statistical indices POD, FAR, CSI and ACC in the comparison between the INSAT-3D-CAPE

with radiosonde CAPE for 20 Stations in India.





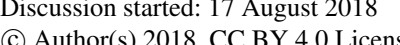

**Fig 5:** Distribution (%) of INSAT-3D, radiosonde and ERA reanalysis CAPE (J kg⁻¹) for all the stations

considered in the study.







**Fig. 6:** The seasonal mean distribution of CAPE (J kg⁻¹) during (a) winter (b) pre-monsoon (c) monsoon and (d) post-monsoon from INSAT-3D data over Indian region.



**Fig. 7:** Frequency distribution (%) of CAPE in (a) weak (b) moderate (c) strong (d) extreme instability

during winter season. (e)-(h): same as (a)-(d) except for pre-monsoon season. (i)-(l): same as

(a)-(d) but for monsoon season and (m)-(p): same as (a)-(d) except for post-monsoon season.



**Fig 8:** Hourly mean distribution of the INSAT-3D CAPE (J kg⁻¹) during the period from 01 April 2014

to 31 March 2017 over the Indian region.





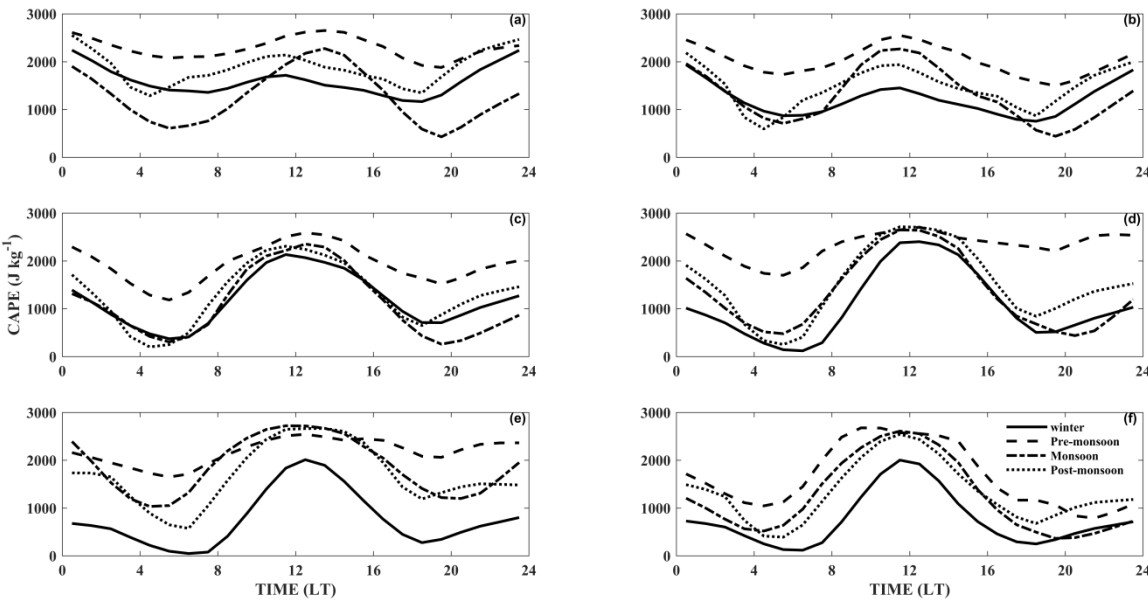

**Fig 9:** Diurnal variation of CAPE (J kg$^{-1}$) over (a) AS (b) BoB (c) SP (d) CI (e) NI (f) NE region of

India during four monsoon seasons.



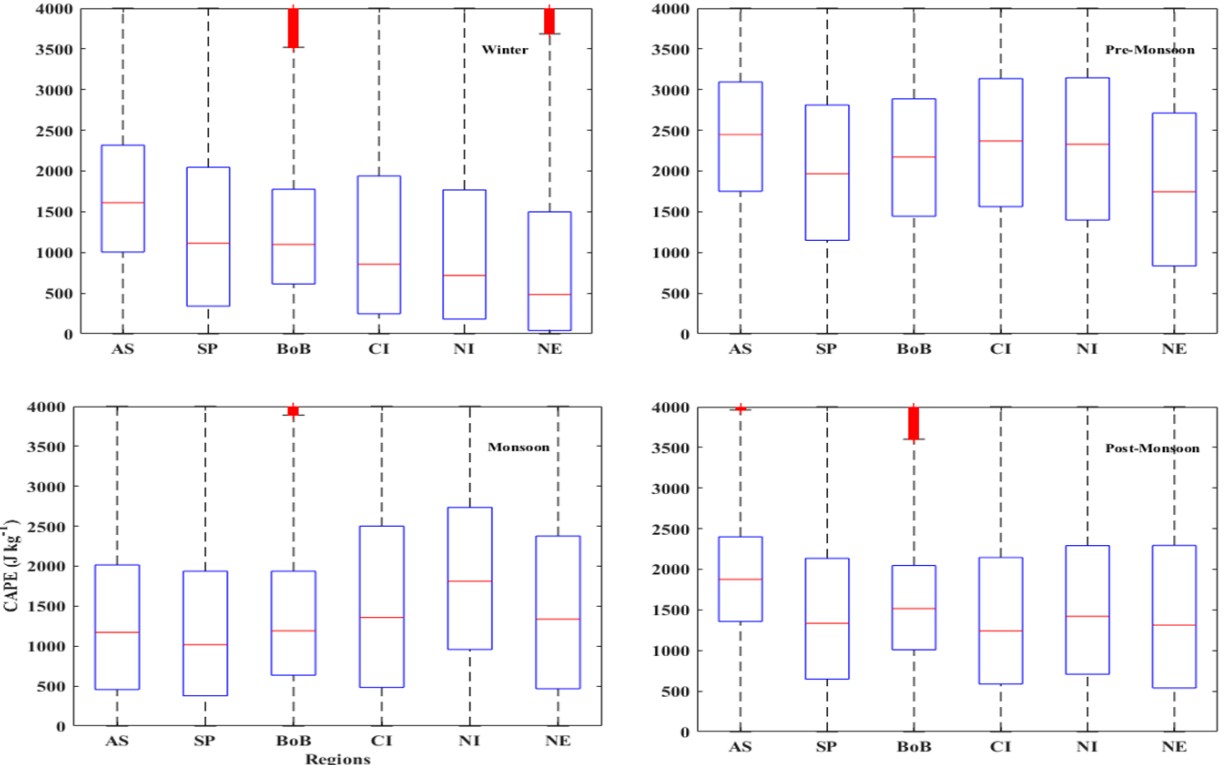

**Fig 10:** Box-plot analysis of distribution of CAPE (J kg$^{-1}$) during (a) winter (b) pre-monsoon (c) monsoon and (d) post-monsoon from INSAT-3D data over the Indian region.
