# Peer review of "Retrieval of convective available potential energy from INSAT-3D measurements: comparison with radiosonde data and its spatial-temporal variations"

_Atmospheric Measurement Techniques, 2018_

## Referee Comment (RC1) · Anonymous Referee #1 · 20 Sep 2018

1. The INSAT-3D is a new satellite and the basic datasets need to validated before CAPE calculation. So I suggest the authors to provide some analysis on how the INSAT temperature, humidity etc. performs over the Indian region, by comparing with radiosonde or reanalysis data. Since India has a large latitudinal extent from near eqautor in South to subtropics in the North, it is essential to investigate whether INSAT data compares well everywhere or there is some spatial inhomogeneity.

2. The authors need to highlight the advantages of their present study i.e. what new can we extract about CAPE by using the INSAT data. The authors mention in abstract

that "In this work, an attempt is made for the first time to estimate CAPE from high spatial and temporal resolution measurements of the INSAT-3D over the Indian region". But there are many other satellites available back from many years and there are several studies related to CAPE over Indian region. So the authors need to discuss the why their work is important and how better it is from the previous estimates.

———————————————————

---

## Referee Comment (RC2) · Anonymous Referee #2 · 21 Sep 2018

Comments on 'Retrieval of convective available potential energy from INSAT-3D measurements: comparison with radiosonde data and its spatial-temporal variations' by Uriya Veerendra Murali Krishna et al., (Atmos. Meas. Tech. Discuss., https://doi.org/10.5194/amt-2018-203)

This paper presents the temporal and spatial distribution of convective available potential energy (CAPE) estimated using INSAT-3D measurements. Initially, these CAPE estimates are compared with that estimated using ERA-Interim reanalysis and the radiosonde measurements obtained from 20 stations that are distributed across India.

[Figure]

Statistical analysis has been made to get confidence on the estimated CAPE values. Finally, the diurnal and seasonal variability in the CAPE is also presented at different geographical locations.

In general, paper is well written and contains significant original contribution. Authors have fully taken advantage of the high spatial and temporal measurements available from INSAT-3D to investigate the diurnal and season variability of CAPE. However, there are few mistakes and sometimes interpretation is missing at some instances without proper literature survey which demands careful editing or re-writing the sentences. Below are the some of the issues which authors need to take care before rendering judgment on the manuscript. Authors are strongly encouraged to revise and re-submit this manuscript.

Major comments:

There are few studies where global measurements of CAPE are available using GPS RO observations (Santhi et al., 2014). Since no observations are there to validate the CAPE at high spatial resolution, small analysis can be made how INSAT-3D estimated CAPE match with GPS RO measured CAPE, particularly over the ocean. Qualitative comparison can also be made.

To the best of my knowledge, INSAT-3D data will not be available during the cloudy times. Since there are two monsoon seasons (SW and NE monsoon) over Indian region, huge data gaps are expected during these two seasons. While making composite analysis at both spatial and temporal scales, results are expected to be biased. How the authors have taken care this issue need to be discussed.

It is mentioned (Line 101) that 'They observed that the INSAT-3D measurements compare better with GPS sonde observations at middle levels (from 900 hPa to 500 hPa).'. In this case how it is going to affect the estimates of CAPE need to be discussed. Further, how large bias observed in water vapor measurements from INSAT-3D is going to affect the CAPE estimates need to be discussed in detail.

[Figure]

Do authors have any explanation why there is a consistent positive (negative) bias (most of the cases) in CAPE values measured by INSAT-3D (ERA-Interim)?

Do authors have any explanation why no variation is seen for large spatial gird in Fig.6 during pre-monsoon?

Introduction is too long without focus. It can be cut to 50% while retaining only relevant information. Diurnal and high spatial resolution studies only need to be highlighted.

Minor comments:

Line 142: It is mentioned that radiosonde data has been taken from University of Wyoming website. Note that this is not the quality checked data. Instead, it will be better to use data from IGRA2.

Line 155: It was mentioned that the resolution of the data utilized is 0.75o×0.75o from ERA-Interim and 0.25oX0.25o from INSAT-3D. How this different spatial resolutions grids are taken care while comparing the CAPE estimates.

Line 282: It is mentioned that 'The estimated CAPE is divided into four categories: weak instability (<500 J kg-1), moderate instability (501-1500 J kg-1), strong instability (1501-3000 J kg-1), and extreme instability (>3000 J kg-1).' Do you have any scientific justification to choose these thresholds? You may provide suitable reference.

Line 338: It is mentioned that 'These regions are: the Arabian Sea (AS; 8-20oN, 65-72oE), Bay of Bengal (BoB; 8-20oN, 80-90oE), South Peninsular India (SP; 8-20oN, 72-80oE), Central India (CI; 20-25oN, 73-82oE), North India (NI; 25-35oN, 73-80oE), and Northeast India (NE; 24-29o34o N, 90-96oE).' Is there any scientific justification to choose these latitude longitude grids? You may provide suitable reference.

Line 411: It is mentioned that several interesting features are noticed. 'The weak instability is predominant during the winter season, the moderate instability is higher during the post-monsoon, the strong instability is more during the monsoon period and the extreme instability is higher during the pre-monsoon months.' Note that these things

are well known to the scientific community.

It seems INSAT-3DR is being launched as a follow up of INSAT-3D. Did you tested how CAPE behaves between these two instruments?

Figure 1 caption: It is better to shift the latitude and longitude along with the name of the station to the running text rather keeping lengthy figure caption.

Figure 8: There are white patches over Tibetan high and also over the central India in few panels. Hope the reasons for the data gaps at these two places is not the same?

Additional references:

Diurnal and long-term variation of instability indices over a tropical region in India R Chakraborty, G Basha, MV Ratnam Atmospheric Research 207, 145-154, 2018

Global morphology of convection indices observed using COSMIC GPS RO satellite measurements YD Santhi, MV Ratnam, SK Dhaka, SV Rao Atmospheric research 137, 205-215, 2014

Diurnal variability of stability indices observed using radiosonde observations over a tropical station: Comparison with microwave radiometer measurements MV Ratnam, YD Santhi, M Rajeevan, SVB Rao Atmospheric Research 124, 21-33, 2013
* * *

---

## Author Comment (AC2) · 21 Nov 2018

The comment was uploaded in the form of a supplement:
https://www.atmos-meas-tech-discuss.net/amt-2018-203/amt-2018-203-AC2-supplement.pdf

---

## Author Response (AR1)

**Response to Reviewer # 1's Comments**

**Comment. 1** *The INSAT-3D is a new satellite and the basic datasets need to validated before CAPE calculation. So I suggest the authors to provide some analysis on how the INSAT temperature, humidity etc. performs over the Indian region, by comparing with radiosonde or reanalysis data. Since India has a large latitudinal extent from near equator in South to subtropics in the North, it is essential to investigate whether INSAT data compares well everywhere or there is some spatial inhomogeneity.*

**Response** **We agree with the referee's suggestion that the Indian region has a large latitudinal extent and spatial inhomogeneity can exists in the retrieval of the satellite data sets.**
**Earlier studies by Mitra et al. (2015) and Ratnam et al. (2016) assessed the temperature and humidity retrievals from the INSAT-3D measurements using GPS sonde, reanalysis and other satellite measurements over the Indian region. They found a good agreement of INSAT-3D retrieved temperature with GPS sonde, reanalysis and satellite estimates below 25°N. The temperature difference was 0.5K with a standard deviation of about 1K, and for humidity, a dry bias (20-30%) was observed between INSAT-3D and GPS sonde data. This is already mentioned in the manuscript.**

**Comment. 2** *The authors need to highlight the advantages of their present study i.e. what new can we extract about CAPE by using the INSAT data. The authors mention in abstract that "In this work, an attempt is made for the first time to estimate CAPE from high spatial and temporal resolution measurements of the INSAT-3D over the Indian region". But there are many other satellites available back from many years and there are several studies related to CAPE over Indian region. So the authors need to discuss the why their work is important and how better it is from the previous estimates.*

**Response** **Several satellites measurements are available which can provide profiles of temperature and water vapour with reasonable accuracies. Most of them are polar orbiting satellites and have limited overpasses especially in the tropics. The other limitation of these polar satellite measurements is poor temporal resolution, even though they have global coverage.**
**The significance of the INSAT-3D is its geostationary orbit, providing the profiles of temperature and water vapour with high temporal (1 hour) and spatial resolution ($0.1^o \times 0.1^o$) over the Indian and the surrounding oceanic regions. In the present study, the authors attempted to calculate CAPE from these high spatial and temporal resolution measurements of INSAT-3D and its performance assessment. To date there are no studies utilizing such high resolution data for such a long period to evaluate and understand the variability of CAPE. Hence, this study provides the direct usability of INSAT-3D data sets in the numerical weather prediction models for now-casting of thunderstorms and for severe weather conditions, which is lacking over the Indian region.**
**The relative sentence has been added in the revised manuscript.**

**Response to Reviewer # 2's Comments**

**General Comment**  *This paper presents the temporal and spatial distribution of convective available potential energy (CAPE) estimated using INSAT-3D measurements. Initially, these CAPE estimates are compared with that estimated using ERA-Interim reanalysis and the radiosonde measurements obtained from 20 stations that are distributed across India. Statistical analysis has been made to get confidence on the estimated CAPE values. Finally, the diurnal and seasonal variability in the CAPE is also presented at different geographical locations.*
*In general, paper is well written and contains significant original contribution. Authors have fully taken advantage of the high spatial and temporal measurements available from INSAT-3D to investigate the diurnal and season variability of CAPE. However, there are few mistakes and sometimes interpretation is missing at some instances without proper literature survey which demands careful editing or re-writing the sentences. Below are the some of the issues which authors need to take care before rendering judgment on the manuscript. Authors are strongly encouraged to revise and re-submit this manuscript.*

**Response**  **We are indebted to the reviewer for his valuable and thoughtful comments on the manuscript. We greatly appreciate the reviewer's time and efforts for evaluating the manuscript. We went through all the referee comments and suggestions and implemented the same in the revised manuscript. Point-to-point clarifications for referee's comments and how we have addressed each recommendation is given below.**

**Comment 1**  *There are few studies where global measurements of CAPE are available using GPS RO observations (Santhi et al., 2014). Since no observations are there to validate the CAPE at high spatial resolution, small analysis can be made how INSAT-3D estimated CAPE match with GPS RO measured CAPE, particularly over the ocean. Qualitative comparison can also be made.*

**Response**  **The GPS RO measurements are very sparse for a particular location and hence are statistically insignificant to compare GPS-RO CAPE with INSAT-3D CAPE. For instance, Santhi et al. (2014) observed a total number of 6 occultations in a month over $2^o \times 2^o$ grid around Gadanki, India. Among these occultation's, only 2 and 4 occultations reached below 0.5 km and 1 km respectively above the surface. When we looked into the number of occultation during the present study period over a particular station ($0.25^o \times 0.25^o$ grid), the total number of occultations are less than 40, which may not be statistically significant. Hence, the reviewer is requested to consider author's proposal for not including the GPS-RO measurements in the revised manuscript.**

**Comment 2**  *To the best of my knowledge, INSAT-3D data will not be available during the cloudy times. Since there are two monsoon seasons (SW and NE monsoon) over Indian region, huge data gaps are expected during these two seasons. While*

*making composite analysis at both spatial and temporal scales, results are expected to be biased. How the authors have taken care this issue need to be discussed.*

**Response**      **We agree with the reviewer. However, for the present analysis we have taken only cloud free conditions. The confidence level for the identification of the cloud free region is based on the cloud flag which is set to zero (CLD_FLG=0).**

**The total number of data for the 20 stations considered in the study during different seasons is provided in Table 1. Since the INSAT-3D data are available for every one hour, the data gaps are not huge during both the monsoon seasons.**

**The relative sentences and Table are added in the revised manuscript.**

*Table 1: The total number of data available in 20 stations during different seasons for the period from April 2014 to March 2017.*

| Station | Latitude | Longitude | Total Number of Data | | | |
|---|---|---|---|---|---|---|
| | | | **Winter** | **Pre-monsoon** | **Monsoon** | **Post-monsoon** |
| Agarthala (AGR) | 23.88 | 91.25 | 3614 | 3272 | 2615 | 2613 |
| Ahmedabad (AHB) | 23.06 | 72.63 | 3808 | 4034 | 3686 | 2891 |
| Amini Divi (AMD) | 11.12 | 72.73 | 3985 | 3776 | 3327 | 2362 |
| Bhuvaneswar (BHU) | 20.25 | 85.83 | 3701 | 3450 | 2389 | 2488 |
| Chennai (CHE) | 13.00 | 80.18 | 3816 | 3712 | 2956 | 1985 |
| Cochin (COC) | 9.95 | 76.26 | 3867 | 3072 | 2905 | 1758 |
| Delhi (DEL) | 28.58 | 77.20 | 2910 | 3316 | 3845 | 2894 |
| Dibrugarh (DIB) | 27.48 | 95.01 | 3287 | 2260 | 2536 | 2595 |
| Gorakhpur (GRK) | 26.75 | 83.36 | 2871 | 3383 | 2769 | 2833 |
| Guwahati (GUW) | 26.10 | 91.58 | 3610 | 3058 | 2956 | 2796 |
| Hyderabad (HYD) | 17.45 | 78.46 | 2550 | 394 | 327 | 1045 |
| Karaikal (KAR) | 10.91 | 79.83 | 3660 | 3458 | 3632 | 1846 |
| Kolkata (KOL) | 22.65 | 88.45 | 3488 | 2924 | 2500 | 2677 |
| Machilipatnam (MAP) | 16.20 | 81.15 | 4001 | 3706 | 2564 | 2322 |

| | | | | | | |
|---|---|---|---|---|---|---|
| Mangalore (MAN) | 12.95 | 74.83 | 4063 | 3542 | 2705 | 2232 |
| Minicoy (MIN) | 8.30 | 73.15 | 3882 | 3346 | 3320 | 2040 |
| Mumbai (MUM) | 19.11 | 72.85 | 4294 | 4321 | 3405 | 2756 |
| Port Blair (PB) | 11.66 | 92.71 | 3676 | 3458 | 2381 | 2178 |
| Trivandrum (TVM) | 8.48 | 76.95 | 3675 | 3116 | 3608 | 1805 |
| Visakhapatnam (VSP) | 17.70 | 83.30 | 3972 | 3747 | 2632 | 2415 |

**Comment 3** *It is mentioned (Line 101) that 'They observed that the INSAT-3D measurements compare better with GPS sonde observations at middle levels (from 900 hPa to 500 hPa).'. In this case how it is going to affect the estimates of CAPE need to be discussed. Further, how large bias observed in water vapor measurements from INSAT-3D is going to affect the CAPE estimates need to be discussed in detail.*

**Response** **CAPE is associated with the changes in the temperature and moisture in the troposphere. However, the changes in CAPE are more related to the moisture present in the boundary layer. Zhang (2002) showed that the net changes in the CAPE come from the thermodynamic changes in the boundary layer. They also showed that the changes in CAPE due to the changes in the temperature and moisture in the free troposphere is about 10% or less when compared to changes in temperature and moisture in the boundary layer. Hence, the changes in the INSAT-3D measurements at the lower levels affect much the changes in the CAPE compared to the changes in the middle levels.**

**The error in estimating the CAPE is determined by applying the standard error propagation formula (Bevington and Robinson, 1992). The error in the CAPE calculation depends on the temperature and water vapour retrievals. In this study, an error of 5% in the measured relative humidity and temperature corresponds to an error of 8% in the calculated CAPE from the INSAT-3D measurements.**

**Comment 4** *Do authors have any explanation why there is a consistent positive (negative) bias (most of the cases) in CAPE values measured by INSAT-3D (ERA-Interim)?*

**Response** **The exact reason is very difficult to be pointed out. However, the reanalysis in general is an assimilated output with prior assumptions. The meteorological parameters of the reanalysis are always underestimated with respect to the observations. For example, Ratnam et al. (2013) have shown the comparison of humidity obtained from SAPHIR–A megha tropiques payload with respect to all reanalysis over the tropics. The humidity was underestimated in reanalysis. This may be the reason for the**

**negative bias in the reanalysis. Radiosonde and INSAT-3D being both observations shows a positive bias.**

| | |
|---|---|
| **Comment 5** | *Do authors have any explanation why no variation is seen for large spatial gird in Fig.6 during pre-monsoon?* |
| **Response** | **In India, the pre-monsoon season is most favourable for the development of thunderstorms over land regions. The surrounding oceanic regions have frequent development of depressions and these depressions sustained over oceanic region for few days. This results in higher values of CAPE over a large spatial extent for few days. This is the reason why no variation in CAPE is observed over large spatial grid during pre-monsoon season.** |
| | |
| **Comment 6** | *Introduction is too long without focus. It can be cut to 50% while retaining only relevant information. Diurnal and high spatial resolution studies only need to be highlighted.* |
| **Response** | **As per the referee's suggestion, the introduction section has been modified in the revised manuscript.** |
| | |
| | **Minor Comments** |
| **Comment 1** | *Line 142: It is mentioned that radiosonde data has been taken from University of Wyoming website. Note that this is not the quality checked data. Instead, it will be better to use data from IGRA2.* |
| **Response** | **We appreciate the referee for his suggestion.** |

The analysis is performed with IGRA2 dataset for the 20 stations considered in the study. As an example, the scatter plot between INSAT-CAPE and IGRA2-CAPE is shown in fig. 1. Even in the case of IGRA2 data also, the INSAT derived CAPE shows good correlation with a correlation coefficient of 0.72.

[Figure]

*Fig. 1: Scatter plot between IGRA2 derived CAPE (J kg$^{-1}$) and INSAT derived CAPE (J kg$^{-1}$) for Chennai during April 2014 to March 2017.*

Similarly, the correlation coefficient for CAPE derived from IGRA2 dataset and INSAT-3D data is provided for the reviewer in table 2. The INSAT-3D derived CAPE shows better correlation for most of the stations considered in the study. This shows the consistency of the INSAT-3D derived CAPE over IGRA2 dataset.

**Table.2:** *Correlation coefficient in the comparison of INSAT-3D CAPE with IGRA2 and Wyoming derived CAPE for 20 stations in India for the period from April 2014 to March 2017.*

| Station | IGRA2 | Radiosonde |
|---------|-------|------------|
| AGR | 0.61 | 0.62 |
| AHB | 0.56 | 0.38 |
| AMD | 0.48 | 0.46 |
| BHU | 0.59 | 0.62 |
| CHE | 0.72 | 0.84 |

| | | |
|---|---|---|
| COC | 0.50 | 0.81 |
| DEL | 0.65 | 0.31 |
| DIB | 0.72 | 0.44 |
| GRK | 0.64 | 0.46 |
| GUW | 0.72 | 0.59 |
| HYD | 0.51 | 0.59 |
| KAR | 0.73 | 0.56 |
| KOL | 0.57 | 0.55 |
| MAP | 0.57 | 0.62 |
| MAN | 0.42 | 0.68 |
| MIN | 0.56 | 0.64 |
| MUM | 0.43 | 0.64 |
| PB | 0.40 | 0.48 |
| TVM | 0.71 | 0.52 |
| VSP | 0.55 | 0.68 |

**It can be observed from the above table that the correlation coefficient in the comparison of INSAT-3D CAPE with IGRA2 and radiosonde CAPE are almost similar for most of the stations. Hence, the inclusion of IGRA2 data in the manuscript is not going to affect the results of the study. So, the reviewer is requested to consider the use of Wyoming university data in the manuscript.**

**Comment 2**  *Line 155: It was mentioned that the resolution of the data utilized is $0.75^o\_0.75^o$ from ERA-Interim and $0.25^oX0.25^o$ from INSAT-3D. How this different spatial resolutions grids are taken care while comparing the CAPE estimates.*

**Response**  **The authors apologize for the typographical error. For the present study, the authors make use of the ERA-Interim data at $0.25^o\times0.25^o$ spatial resolution.**
**The relative sentence is modified in the manuscript.**

**Comment 3**  *Line 282: It is mentioned that 'The estimated CAPE is divided into four categories: weak instability (<500 J kg-1), moderate instability (501-1500 J kg-1), strong instability (1501-3000 J kg-1), and extreme instability (>3000 J kg-1).' Do you have any scientific justification to choose these thresholds? You may provide suitable reference.*

**Response**  **These CAPE ranges are considered arbitrarily.**

**Comment 4**  *Line 338: It is mentioned that 'These regions are: the Arabian Sea (AS; 8-20oN, 65-72oE), Bay of Bengal (BoB; 8-20oN, 80-90oE), South Peninsular India (SP; 8-20oN, 72-80oE), Central India (CI; 20-25oN, 73-82oE), North India (NI; 25-35oN, 73-80oE), and Northeast India (NE; 24-29o34o N, 90-96oE).' Is there any scientific justification to choose these latitude longitude grids? You may provide suitable reference.*

| | |
|---|---|
| **Response** | **To understand the diurnal variation of CAPE over different parts of India, the study region is divided (latitude-wise for uniformity) into six sub-regions; Arabian Sea (AS; 8-20$^o$N, 65-72$^o$E), Bay of Bengal (BoB; 8-20$^o$N, 80-90$^o$E), South Peninsular India (SP; 8-20$^o$N, 72-80$^o$E), Central India (CI; 20-25$^o$N, 73-82$^o$E), North India (NI; 25-35$^o$N, 73-80$^o$E), and Northeast India (NE; 24-29$^o$N, 90-96$^o$E) as given in Raut et al. (2009). The relative reference is added in the text in the revised manuscript.** |
| **Comment 5** | *Line 411: It is mentioned that several interesting features are noticed. 'The weak instability is predominant during the winter season, the moderate instability is higher during the post-monsoon, the strong instability is more during the monsoon period and the extreme instability is higher during the pre-monsoon months.' Note that these things are well known to the scientific community.* |
| **Response** | **The above sentence is re-written as "The spatial and temporal distribution of CAPE reveals that the weak instability is predominant during winter season, moderate instability is higher during post-monsoon, strong instability is more during monsoon period and extreme instability is higher during pre-monsoon months".** |
| **Comment 6** | *It seems INSAT-3DR is being launched as a follow up of INSAT-3D. Did you tested how CAPE behaves between these two instruments?* |
| **Response** | **INSAT-3DR is a redundancy payload for INSAT-3D and hence we have not attempted to calculate CAPE with INSAT-3DR data. In future, this will be our follow-up study.** |
| **Comment 7** | *Figure 1 caption: It is better to shift the latitude and longitude along with the name of the station to the running text rather keeping lengthy figure caption.* |
| **Response** | **Referee's suggestion is implemented in the manuscript.** |
| **Comment 8** | *Figure 8: There are white patches over Tibetan high and also over the central India in few panels. Hope the reasons for the data gaps at these two places is not the same?* |
| **Response** | **The white patches in the Tibetan high are due to the non-availability of data due to topography. Over Central India, it is mostly due to non-availability of data.** |

**Refrences:**

Venkat Ratnam, M., Basha, G., Krishna Murthy, B. V., & Jayaraman, A. (2013). Relative humidity distribution from SAPHIR experiment on board Megha-Tropiques satellite mission: Comparison with global radiosonde and other satellite and reanalysis datasets, *Journal of Geophysical Research Atmospheres*, 18, 1–9, doi:10.1002/jgrd.50699.

Raut, B. A., Karekar, R. N., & Puranik, D. M. (2009). Spatial distribution and diurnal variation of cumuliform clouds during Indian Summer Monsoon. *Journal of Geophysical Research Atmospheres*, *114*(11), 1–12. https://doi.org/10.1029/2008JD011153

Santhi, Y. D., Ratnam, M. V., Dhaka, S. K., & Rao, S. V. (2014). Global morphology of convection indices observed using COSMIC GPS RO satellite measurements. *Atmospheric Research*, *137*, 205–215. https://doi.org/10.1016/j.atmosres.2013.10.002

Zhang, G. J. (2002). Convective quasi-equilibrium in midlatitude continental environment and its effect on convective parameterization. *Journal of Geophysical Research Atmospheres*, *107*(14), 1–16. https://doi.org/10.1029/2001JD001005

[revised manuscript text omitted]
. Santhi et al. (2014) estimated various stability indices using COSMIC GPS--RO profiles and the uncertainty in estimating these stability indices. They also studied the diurnal variation of these stability indices over Gadanki, India. In order to study the diurnal variation of stability indices, they integrated the data over a season as the occultations were sparse and hence not adequate to study on daily scale. This limitation of under sampling can be overcome by the use of geostationary satellites. These geostationary satellite measurements provide near-continuous monitoring of  the atmosphere with better spatial coverage, which is helpful in nowcasting of convection (Koenig and de Coning, 2009). Siewert et al. (2010) discussed the advantages of the METEOSAT Second Generation (MSG) system in deriving the instability indices and to predict the convection initiation over the Central Europe and South Africa.

Using the MSG satellite measurements, de Coning et al. (2011) derived a new convection indicator, the combined instability index which can calculate the probability of convection over the South Africa. They showed that the combined instability index can predict the convection better than the individual instability indices like K-index, total totals index etc.

Jewett and Mecikalski (2010) developed an algorithm to derive convective momentum fluxes from the Geostationary Operational Environmental Satellite (GOES) measurements. The advantage of this algorithm is that it can be used in any convective environment.

Recently, the Indian Space Research Organization (ISRO) launched the Indian

National Satellite System (INSAT-3D), which is a geostationary satellite that provides the profile of temperature and relative humidity with high temporal and spatial resolution.

Several researchers evaluated the temperature and relative humidity measurements from the INSAT-3D. Mitra et al. (2015) evaluated the INSAT-3D temperature and moisture retrievals up to 100 hPa with GPS sonde observations for the period January-

May 2014. They observed that the INSAT-3D measurements compare better with GPS sonde observations at middle levels (from 900 hPa to 500 hPa). The assessment of the quality of temperature and water vapour obtained from the INSAT-3D with in-situ, satellite, and reanalysis datasets by Ratnam et al. (2016) revealed that the INSAT-3D measurements agree well with the GPS sonde observations,  satellite measurements and reanalysis datasets below $25^{\circ}$N. The temperature difference was 0.5K with a standard deviation of about 1K, and for humidity, a dry bias (20-30%) was observed between INSAT-3D and other satellite measurements and reanalysis data. Hence, these satellite measurements also suffer from some inherent shortcomings and have biases and random errors. Therefore, it is essential to evaluate the satellite products with conventional measurements to quantify the direct usability of these products.

The objective of the present study is to calculate CAPE from high spatial and temporal resolution measurements of INSAT-3D over the Indian region and its performance assessment. To date there are no studies utilizing such high-resolution data for such a long period to evaluate and understand the variability of CAPE. Hence, this study provides the direct usability of INSAT-3D data sets in the numerical weather prediction models for nowcasting of thunderstorms and for severe weather conditions, which is lacking over the Indian region. In this work, we first attempted 
[revised manuscript text omitted]

The error in estimating the CAPE is determined by applying the standard error propagation formula (Bevington and Robinson, 1992). The error in the CAPE calculation depends on the temperature and water vapour retrievals. In this study, an error of 5% in the measured relative humidity and temperature corresponds to an error of 8% in the calculated CAPE from the INSAT-3D measurements.

**4. Results and Discussions**

Three years of data collected from the INSAT-3D measurements are utilized to estimate CAPE over the Indian region. These estimates are compared with the radiosonde derived CAPE at 0000 and 1200 UTC along with the ERA-Interim reanalysis CAPE data.

**4.1. *Comparison of INSAT-3D CAPE with radiosonde and ERA-Interim estimated CAPE with radiosonde measurements**

In this work, 20 stations are selected for which the radiosonde profiles are available during the study period (location shown in Fig. 1). Table 1 provide the details of the stations considered along with the total number of data available from INSAT-3D measurements during four seasons (winter: December to February; pre-monsoon: March to May; monsoon: June to September; and post-monsoon: October and November)Table 1 provide the details of the stations considered along with the total number of data available during different monsoon seasons (winter; December to February, pre-monsoon;; March to May, monsoon; June to September, and post-monsoon; October and November). Fig. 2 shows the correlation coefficient along with the number of data points used in the analysis for the comparison between the estimated CAPE from the INSAT-3D measurements, ERA-Interim CAPE, and radiosonde derived CAPE. Here, the comparison is performed only when the INSAT-3D and radiosonde CAPE as well as INSAT-3D and ERA-Interim CAPE greater than zero. From this figure, it is noticed that the INSAT-3D estimated CAPE shows the better correlation with radiosonde CAPE compared to the ERA-Interim CAPE. In general, the coastal stations show a higher correlation coefficient than that of the other stations for the INSAT-3D estimated

CAPE. All the coastal stations show a correlation higher than 0.6 except Trivandrum. The correlation coefficient is as high as 0.84 for Chennai among all the stations. The correlation values are lower for the stations located near the foothills of  Himalayas and north-east (NE) regions of India. A weak correlation of 0.31 is found for Delhi. For the ERA-Interim data, INSAT-3D CAPE shows less correlation coefficient for all the stations except Amini

Divi compared to radiosonde CAPE

.

Even in the ERA-Interim CAPE, the coastal stations show better correlation compared to other stations. The ERA-Interim CAPE shows a higher correlation for Minicoy and the correlation is minimum for Delhi. This result suggests that the INSAT-3D

CAPE measurements agree well with the radiosonde measurements.

To elucidate the consistency of INSAT-3D estimated CAPE against the ERA-Interim

[revised manuscript text omitted]

is higher for lower values $(< 1000$ J kg$^{-1})$ in all the three datasets (radiosonde, INSAT-3D and ERA-Interim) for all the stations.

The INSAT-3D and ERA-Interim estimated CAPE is higher than the radiosonde measurements in the lower ranges. The INSAT-3D estimated CAPE matches well with the radiosonde measurements above ~3000 J kg$^{-1}$.

The spatial and temporal distribution of CAPE reveals that the weak instability is predominant during  winter season,  moderate instability is higher during  post-monsoon,  strong instability is more during  monsoon period and  extreme instability is higher during  pre-monsoon months. The diurnal variation in mean CAPE shows a bimodal distribution with  primary peak around mid-night and secondary peak in the afternoon times for most of the regions and in different seasons. The seasonal mean CAPE shows that the land areas show lower CAPE during  winter, whereas the oceans show the highest CAPE during  pre-monsoon season. The higher values of

CAPE over the oceanic regions may be due to  higher occurrence of the tropical cyclones during the pre-monsoon season. Further, the north- western parts of India  show higher CAPE among other land regions. Overall, the

INSAT-3D estimated CAPE is in close agreement with the radiosonde derived CAPE. As the

INSAT-3D provides high temporal and spatial resolution data, hence it can be used for now- casting and severe weather warnings in the numerical prediction models.

**Acknowledgement**

The authors sincerely acknowledge the Director, IITM for his constant support and encouragement during this study. The authors would like to acknowledge Meteorological and

Oceanographic Satellite Data Archival Centre (MOSDAC) of Space Application Centre (SAC), ISRO for supplying the INSAT-3D data. The authors are grateful to the

Department of Atmospheric Science, University of Wyoming for access to their radiosonde data archive. The authors acknowledge  the ERA-Interim team for providing their data. All the dataset are freely available and can be downloaded from their respective archive. The authors would like to sincerely thank the Editor and two anonymous reviewers for his their insightful comments that improved the quality of the manuscript.

**Table Captions:**

**Table 1:** The total number of data available during four  seasons for the 20 stations for the period from April 2014 to March 2017.

**Table 2:** Contingency table for the comparison of INSAT-3D estimated CAPE with radiosonde measured CAPE. The CAPE threshold is  considered as 0 J kg$^{-1}$.

**Figure Captions:**

**Fig 1:** The geographical map of India representing the 20 stations considered for the present study.

**Fig 2: (a)** Correlation  coefficient in the comparison of INSAT-3D

CAPE with radiosonde and ERA-Interim CAPE for 20 stations in India for the  period April 2014 to March 2017. **(b)** Number of data points in the comparison of INSAT-3D CAPE with radiosonde and ERA-Interim

CAPE.

**Fig 3:** Bias in the comparison between INSAT-3D- CAPE and ERA-Interim reanalysis CAPE with radiosonde for different  stations in India for the  period April, 2014 – March, 2017.

**Fig 4:** Statistical indices (a) POD, (b) FAR, (c) CSI and (d) ACC in the comparison between the INSAT-3D- CAPE with radiosonde CAPE for 20  stations in India.

**Fig 5:** Distribution (%) of INSAT-3D, radiosonde and ERA-Interim reanalysis CAPE (J kg$^{-1}$) for all the stations considered in the study.

**Fig. 6:** The  distribution of CAPE (J kg$^{-1}$) during (a) winter (b) pre-monsoon (c) monsoon and (d) post-monsoon from INSAT-3D data over Indian region.

**Fig. 7:** Frequency distribution (%) of CAPE in (a) weak (b) moderate (c) strong (d) extreme instability during winter season. (e)-(h): same as (a)-(d) except for pre-monsoon season. (i)-(l): same as (a)-(d) but for monsoon season and (m)-(p): same as (a)-(d) except for post-monsoon season.

**Fig 8:** Hourly mean distribution of  INSAT-3D CAPE (J kg$^{-1}$) during the period from 01 April 2014 to 31 March 2017 over the Indian region.

**Fig 9:** Diurnal variation of CAPE (J kg$^{-1}$) over (a) AS (b) BoB (c) SP (d) CI (e) NI (f) NE region of India during four  seasons.

**Fig 10:** Box-plot analysis  of CAPE (J kg$^{-1}$) during (a) winter (b) pre-monsoon (c) monsoon and (d) post-monsoon from INSAT-3D data over six sub-regions over the Indian sub-continent.

**Table 1:** The total number of data available during four  seasons for the 20 stations for the period from April, 2014 to March, 2017.

| Station | Latitude | Longitude | Total Number of Data | | | |
|---|---|---|---|---|---|---|
| | | | Winter | Pre-monsoon | Monsoon | Post-monsoon |
| Agarthala (AGR) | 23.88 | 91.25 | 3614 | 3272 | 2615 | 2613 |
| Ahmedabad (AHB) | 23.06 | 72.63 | 3808 | 4034 | 3686 | 2891 |
| Amini Divi (AMD) | 11.12 | 72.73 | 3985 | 3776 | 3327 | 2362 |
| Bhuvaneswar (BHU) | 20.25 | 85.83 | 3701 | 3450 | 2389 | 2488 |
| Chennai (CHE) | 13.00 | 80.18 | 3816 | 3712 | 2956 | 1985 |
| Cochin (COC) | 9.95 | 76.26 | 3867 | 3072 | 2905 | 1758 |
| Delhi (DEL) | 28.58 | 77.20 | 2910 | 3316 | 3845 | 2894 |
| Dibrugarh (DIB) | 27.48 | 95.01 | 3287 | 2260 | 2536 | 2595 |
| Gorakhpur (GRK) | 26.75 | 83.36 | 2871 | 3383 | 2769 | 2833 |
| Guwahati (GUW) | 26.10 | 91.58 | 3610 | 3058 | 2956 | 2796 |
| Hyderabad (HYD) | 17.45 | 78.46 | 2550 | 394 | 327 | 1045 |
| Karaikal (KAR) | 10.91 | 79.83 | 3660 | 3458 | 3632 | 1846 |
| Kolkata (KOL) | 22.65 | 88.45 | 3488 | 2924 | 2500 | 2677 |
| Machilipatnam (MAP) | 16.20 | 81.15 | 4001 | 3706 | 2564 | 2322 |
| Mangalore (MAN) | 12.95 | 74.83 | 4063 | 3542 | 2705 | 2232 |
| Minicoy (MIN) | 8.30 | 73.15 | 3882 | 3346 | 3320 | 2040 |

| | | | | | |
|---|---|---|---|---|---|
| Mumbai (MUM) | 19.11 | 72.85 | 4294 | 4321 | 3405 | 2756 |
| Port Blair (PB) | 11.66 | 92.71 | 3676 | 3458 | 2381 | 2178 |
| Trivandrum (TVM) | 8.48 | 76.95 | 3675 | 3116 | 3608 | 1805 |
| Visakhapatnam (VSP) | 17.70 | 83.30 | 3972 | 3747 | 2632 | 2415 |

**Table 2:** Contingency table for the comparison of INSAT-3D estimated CAPE with radiosonde measured CAPE. The CAPE threshold is  considered as 0 J kg$^{-1}$.

|  | Radiosonde ≥Threshold | Radiosonde < Threshold |
|---|---|---|
| Satellite ≥Threshold | Hits (a) | False alarms (b) |
| Satellite <Threshold | Misses (c) | Correct negatives (d) |

[Figure]

**Fig 1:** The geographical map of India representing the 20 stations considered for the present study.

[Figure]

[Figure]

**Fig 2: (a)** Correlation coefficient in the comparison of INSAT-3D CAPE with radiosonde and ERA-Interim CAPE       for 20 stations in India for the  period April 2014 to March 2017. **(b)** Number of data points in the comparison of INSAT-3D- CAPE, ERA CAPE with radiosonde and ERA-Interim

CAPE.

[Figure]

**Fig 3:** Bias in the comparison between INSAT-3D- CAPE and ERA-Interim reanalysis

CAPE with radiosonde for different Stations in India for the Period period April,

– March, 2017.

[Figure]

**Fig 4:** Statistical indices (a) POD, (b) FAR, (c) CSI and (d) ACC in the comparison between the INSAT-3D- CAPE with radiosonde CAPE for 20 Stations stations in India.

[Figure]

**Fig 5:** Distribution (%) of INSAT-3D, radiosonde and ERA-Interim reanalysis CAPE (J kg⁻¹)

for all the stations considered in the study.

[Figure]

**Fig. 6:** The  distribution of CAPE (J kg⁻¹) during (a) winter (b) pre-monsoon (c) monsoon and (d) post-monsoon from INSAT-3D data over Indian region.

[Figure]

**Fig. 7:** Frequency distribution (%) of CAPE in (a) weak (b) moderate (c) strong (d) extreme instability during winter season. (e)-(h): same as (a)-(d) except for pre-monsoon season. (i)-(l): same as (a)-(d) but for monsoon season and (m)-(p): same as (a)-(d)

except for post-monsoon season.

[Figure]

**Fig 8:** Hourly mean distribution of  INSAT-3D CAPE (J kg⁻¹) during the period from 01 April 2014 to 31 March 2017 over the Indian region.

[Figure]

**Fig 9:** Diurnal variation of CAPE (J kg⁻¹) over (a) AS (b) BoB (c) SP (d) CI (e) NI (f) NE

region of India during four  seasons.

[Figure]

**Fig 10:** Box-plot analysis  of CAPE (J kg$^{-1}$) during (a) winter (b) pre-monsoon
(c) monsoon and (d) post-monsoon from INSAT-3D data over six sub-regions over the Indian
sub-continent.